# Traffic Noise at Moderate Levels Affects Cognitive Performance: Do Distance-Induced Temporal Changes Matter?

**DOI:** 10.3390/ijerph20053798

**Published:** 2023-02-21

**Authors:** Leon Müller, Jens Forssén, Wolfgang Kropp

**Affiliations:** Division of Applied Acoustics, Chalmers University of Technology, 41296 Gothenburg, Sweden

**Keywords:** traffic noise, cognitive performance, perceived workload, continuous performance test, sustained attention, inhibitory control, NASA-TLX, noise preference

## Abstract

Urbanization leads to an increased demand for urban housing, which can be met by building dwellings closer to streets. Regulations often limit equivalent sound pressure levels which do not account for changes in time structure that occur when decreasing the road distance. This study investigates the effect of such temporal changes on subjective workload and cognitive performance. A group of 42 participants performed a continuous performance test as well as a NASA-TLX workload test under three different sound conditions, i.e., close traffic, far traffic, both with the same equivalent sound pressure level of $L_{\mathrm{Aeq}}\approx 40$ dB, and silence. Additionally, participants answered a questionnaire regarding their preferred acoustic environment for concentrated working. Significant effects of the sound condition on the multivariate workload results as well as on the number of commission errors in the continuous performance test were found. Post hoc tests showed no significant differences between the two noise conditions, but there were significant differences between noise and silence. This indicates that moderate traffic noise levels can influence cognitive performance and perceived workload. If there is a difference in the human response to road traffic noise with constant $L_{\mathrm{Aeq}}$ but different time structures, the used methods are not suitable to detect them.

## 1. Introduction

Research investigating the impact of noise on humans already started in the early 20th century [1]. Motivated by the progressing industrialization, these experiments often focused on workers being exposed to high machine noise levels which were found to affect humans both on a biophysical as well as a cognitive level [2]. In the following decades, more and more research on the human response to noise was conducted, and it became clear that noise exposure does not only have temporary effects [3,4] but can cause long-term damage to the auditory system as well as influence health and well-being. These effects include hypertension [5,6], cardiovascular diseases [7,8], sleeping disorders [9,10], learning impairments [11,12], annoyance [13,14] and other health problems [15,16,17]. In addition to occupational noise exposure, environmental and especially road-traffic noise has a major impact on our society and is, according to the European Environment Agency, responsible for thousands of premature deaths as well as millions of cases of chronic high annoyance and sleep disturbance in the European territory [18].

While local governments in collaboration with organizations such as the EU or the World Health Organization aim to reduce overall noise exposure by regulating traffic noise, such regulations come at the price of either reduced traffic flow, increased construction costs or less available space for dwellings. The latter is especially relevant for countries with a shortage of affordable urban housing such as Sweden [19], which therefore decided to lower the housing standards for small apartments, allowing higher noise levels [20,21]. A similar trend can be observed in the Netherlands which, with the expected implementation of a new environment and planning act, might allow increased façade levels of up to 70 dB [22,23]. As a result of these changes, housing can be built closer to streets. Regulations often limit time-averaged energy-related measures such as equivalent sound pressure levels ($L_{\mathrm{Aeq}}$), which can be reduced by increasing the façade insulation or setting up noise screens. However, reducing the distance between a road and a building while only limiting time-averaged sound pressure levels also leads to a change in traffic noise time structure as visualized in Figure 1. Due to these temporal changes, traffic noise from loud vehicles far away is perceived differently than noise from quieter cars close by, even though the $L_{\mathrm{Aeq}}$ at a listening position might be identical. While research showed that factors such as noise level [24] and frequency composition [25] may be of relevance for the human response, it is currently unclear whether these temporal differences matter.

Schaeffer et al. [26] performed a study investigating the effect of macro-temporal changes in traffic noise on annoyance and cognitive performance, and they found that while shorter but regular breaks between individual vehicle passages were perceived as less annoying than longer but irregular breaks, cognitive performance as assessed in an attention-based Stroop task [27] did not vary with noise time structure. The differences in macro-temporal traffic noise distribution of this study were implemented by varying the duration of quiet periods between different vehicle pass-by events while maintaining the overall $L_{\mathrm{Aeq}}$. Those modifications correspond to a variation of traffic flow in conjunction with a change in the sound pressure level of individual passages and hence are different from the time structural changes caused by reducing the physical distance to a street. Nevertheless, these findings might indicate that a scenario of distant traffic and hence no pronounced quiet periods could be perceived as more annoying than close traffic with more pronounced quiet periods and that there might be no difference in cognitive performance.

A similar study was performed by Lavandier et al. [28], who investigated the influence of fluctuating road traffic noise on perceived disturbance and performance in a reading task. Thereby, the authors found that the text comprehension rate decreases with increasing noise levels and that the reading speed during the rising front of a traffic noise peak decreases compared to a constant background noise level. This outcome might indicate that close traffic noise, which is more dynamic and hence has more pronounced peaks, could affect reading speed more than distant traffic noise with a more constant time structure.

The results of these studies, in combination with everyday observations such as that irrelevant speech in an office, footsteps from neighbors or the sound of a dripping water tap can be perceived as stressful or distracting even though those signals exhibit low sound pressure levels, show that not only acoustic energy but also informational content matters for the human perception. This means that energy-related measures alone might not be sufficient to describe the human response to traffic noise, which motivates this study to evaluate the effect of distance-induced temporal changes on cognitive performance and perceived workload.

## 2. Methods

In this study, a specially designed laboratory was used to realistically reproduce indoor road-traffic noise scenarios with different road-to-façade distances but the same $L_{\mathrm{Aeq}}$. An experiment consisting of two sub-tasks was conducted, measuring sustained attention and inhibitory control as well as perceived workload under three different sound conditions. Thereby, the working hypothesis was that the presence of traffic noise affects performance in both tasks and that noise from nearby traffic has a higher impact than noise from distant traffic. Additionally, the participants answered a questionnaire regarding their preferred acoustic environment for concentrated working.

### 2.1. Auralization

The aim of the auralization method used in this study was to create perceptually plausible stimuli in a natural environment. Therefore, an acoustic transmission suite was modified to a so-called Living Room Lab, as shown in Figure 2a. This lab consists of a listening room furnished to match the look and feel of a regular living room, as shown in Figure 2b, as well as a transmitting room equipped with a linear loudspeaker array consisting of 24 studio loudspeakers as well as two subwoofers, as shown in Figure 2c. Both rooms are acoustically isolated from the environment and separated from each other via a gypsum double wall with an aluminum double glass window. The fundamental assumption behind the auralization approach is that the majority of sound emitted by an outdoor source propagates through the window; hence, a correct reproduction of the sound field on the outside of the window leads to a spatially correct sound field in the entire listening room. Wave field synthesis [29] was implemented to reproduce the outdoor sound field of a moving vehicle including propagation effects such as source directivity, ground reflections and Doppler effect. As source signals, multichannel semi-anechoic heavy vehicle recordings were used. During the experiments, the window was covered by an acoustically transparent curtain in order to hide the speakers from the participants. The accuracy of the used auralization method was evaluated in [30].

### 2.2. Stimuli

Three different sound conditions were evaluated: total silence (Silence ), traffic at 10 m road-to-façade distance (Close) and traffic at 50 m road-to-façade distance (Far), both with 600 heavy vehicle passages per hour. Thereby, the speed of the vehicles was set to 50 km/h with random variations of ±10 km/h for both traffic simulations assuming an exponential traffic flow distribution [31]. The overall equivalent sound pressure level of both traffic noise signals was normalized to $L_{\mathrm{Aeq}}\approx 40$ dBA at the indoor participant position which, according to [32,33], could be classified as a moderate noise level. Figure 3 shows an excerpt of the time structure as well as third-octave band levels for all three stimuli. Table 1 summarizes the properties of all three stimuli including sound pressure levels measured at the listening position without a participant being present in the room. Binaural stimuli recordings are openly available in https://doi.org/10.5281/zenodo.7565815 (accessed on 25 January 2023).

### 2.3. Experimental Procedure

The experiment was performed in December 2021 at the facilities of the Division of Applied Acoustics, Chalmers University of Technology, Gothenburg, Sweden. The overall experiment procedure is shown in Figure 4. All participants started the experiment by filling out a noise preference questionnaire (cf. Section 2.3.1) followed by a continuous performance test (CPT) of sustained attention and response inhibition (cf. Section 2.3.2) under the first sound condition. After the first CPT round, the subjects answered a NASA-TLX questionnaire (cf. Section 2.3.3) assessing the perceived task load of the previously performed CPT. After a 10-min break, an additional CPT round was performed under the second sound condition followed by another break and a final CPT round under the third sound condition. In order to avoid practice or fatigue effects, the order of the three sound conditions Close, Far and Silence for each CPT round was fully balanced among all 42 participants, i.e., all six possible order permutations were performed by seven participants each. All participants performed the experiment individually, and the overall experiment duration was approximately 70 min per participant.

#### 2.3.1. Noise Preference Questionnaire

The first step for each participant was to answer a questionnaire regarding their preferred acoustic environment for concentrated working. Thereby, the subjects were asked to freely describe sounds that they prefer as well as sounds that they find disturbing when focusing on a task. Additionally, the participants rated their loudness preference for concentrated working on a discrete 21-point scale ranging from “Quiet” (0), i.e., the total absence of sound, to “Loud” (20), i.e., their personal highest bearable sound pressure level.

#### 2.3.2. Continuous Performance Test

A continuous performance test (CPT) traditionally assesses sustained attention often by using a simple computerized paradigm [34]. The test performed in this study resembles the commercially available Conners CPT [35], which is a popular tool for the assessment of attention and inhibitory control [36,37] such as in the diagnosis of attention-deficit/hyperactivity disorder (ADHD) [38]. To our best knowledge, this paradigm has not been used yet to investigate the effects of complex traffic noise on cognitive performance.

During the experiment, the participants were sequentially presented with single letters on a computer monitor and instructed to press a button for all letters except “X”, which constituted 10% of the total set of letters. In one CPT round, a sequence of 360 letters was organized in six cycles which each consisted of 3 blocks of 20 letters, as visualized in Figure 4. Thereby, each block of letters was randomly presented with either 1 s, 2 s or 4 s interstimulus interval (ISI) and a presentation time of 250 ms, resulting in an overall test duration of 14 min per round. The high target-to-distractor ratio of this paradigm means that the participant has to respond frequently and inhibit responding when the rare non-target “X” shows up. While CPTs are commonly referred to as tests of sustained attention, research such as [39] indicates that due to the high target-to-distractor ratio, this response-inhibition CPT variant is rather an indicator for inhibitory control which describes the suppression of behavioral responses to goal-irrelevant stimuli and is a core executive function of the cognitive system [40,41].

In a diagnostic context, different measures of the CPT would be compared to normative data in order to assess the overall performance of a subject. However, in this study, only the overall number of commission errors—i.e., cases where the participant responded even though the non-target letter “X” appeared, as well as the mean response time, i.e., the time it took for a participant to press the button after a target letter was shown—was analyzed regarding within-subject effects between the three evaluated sound conditions. Thereby, a high number of commission errors can indicate both inattention or impulsivity, while a fast response time indicates impulsivity and a slow response time indicates inattention [34].

The CPT test as well as the synchronized traffic noise playback was implemented in MATLAB R2021b, and the response time was measured with sub-millisecond accuracy using a micro-controller connected to a response button and a photo-sensor mounted to the computer monitor.

#### 2.3.3. NASA-TLX Test

In order to investigate the subjective task load under different sound conditions, a NASA-TLX task load test was performed. This test was first published by Hart and Staveland in 1988 [42] and has since then become a standard self-reported questionnaire to evaluate perceived workload [43] which has also been used in the context of acoustic research [44,45,46]. After completing a task, the NASA-TLX test asks a subject to rate the perceived Mental Demand, Physical Demand, Temporal Demand, Performance, Effort and Frustration each on a scale ranging from 0 to 100 in 5-point steps. Thereby, the scale for Performance is labeled from “Perfect” (0) to “Failure” (100) and all other subscales range from “Very Low” (0) to “Very High” (100). The original NASA-TLX test also contains a weighting of those individual dimensions in order to calculate an overall workload score. However, since it was expected that the impact of traffic noise might vary between the different task load metrics, it was decided to use a common modification of the NASA-TLX test by skipping this additional step and evaluating the results of the subscale ratings individually.

In order to limit the overall experiment duration, each participant performed only one NASA-TLX test for a single sound condition after the first CPT round (see Figure 4), i.e., the task load was assessed using independent measures. Due to the fully balanced experiment design, this resulted in 14 participants performing the TLX test for each sound condition.

### 2.4. Participants

The experiment was performed by 42 participants (16 females and 26 males) who were mostly recruited from Chalmers students and faculty members. The participants were aged between 20 and 45 years old with a median age of 26 years, and the majority had an educational background in acoustics. All participants had self-reported normal hearing and gave their written consent for participation as well as collection and processing of their personal data.

## 3. Results

All results were analyzed in MATLAB R2022b and SPSS 28.0.1.0; effects were considered significant at the 0.05 level unless noted otherwise. The data presented in this study are openly available in https://doi.org/10.5281/zenodo.7565815 (accessed on 25 January 2023).

### 3.1. Noise Preference

The free text questionnaire answers regarding positive and negative acoustic environments for concentrated working were assigned to 24 different sound categories and analyzed regarding the number of responses for each category, as shown in Figure 5a. These results show that most subjects of the evaluated group prefer calm music and silence as acoustic working environments, and the dominating sound category for negative acoustic environments is speech.

The results from the preferred loudness shown in Figure 5b indicate that the majority of participants stated to prefer a relatively quiet environment (M=5.048, SD=2.811); only four out of 42 participants stated preferring absolute silence. Additionally, female participants reported to prefer lower loudness (M=4.125, SD=2.729) than males (M=5.615, SD=2.758). However, this difference was found to be not significant, as assessed by a one-way analysis of variance (ANOVA; F(1,40)=2.915, p=0.096, partial $\eta^{2}=0.122$).

### 3.2. Perceived Task Load

#### 3.2.1. Analysis

The impact of the sound condition on the NASA-TLX results presented in Figure 6 was analyzed using a one-way multivariate analysis of variance (MANOVA) with Wilk’s Λ as a multivariate test statistic. An explorative data analysis showed that the results for the parameter Physical Demand contain extreme univariate outliers and are not normally distributed; hence, it was decided to exclude this subscale from the MANOVA. Prior to the analysis, it was confirmed that the data fulfill all assumptions for a MANOVA except homogeneity of error variances for the Mental Demand results as assessed by Levene’s test (p=0.015). This violation of the homogeneity assumption was considered in the choice of Wilk’s Λ as a multivariate test statistic [47] as well as Welch ANOVAs and the Games–Howell multiple comparisons method as post hoc test.

Every participant performed only one NASA-TLX test which, even though female participants were equally distributed among the resulting three groups, led to only five to six female subjects per sound condition. This was found to be not sufficient for a meaningful statistic analysis; hence, it was decided to not include gender as a between-subject factor in the MANOVA.

#### 3.2.2. Results

Performing a one-way MANOVA revealed a significant effect of the sound condition on the combined NASA-TLX results for Mental Demand, Temporal Demand, Performance, Effort and Frustration (F(10,70)=2.394, p=0.017, partial $\eta^{2}=0.255$, Wilk’s Λ=0.555) with a large effect size [48]. Post hoc Welch ANOVAs were conducted for every evaluated dependent variable. The results presented in Table 2 show a significant effect of the sound condition on Mental Demand but not on the other task load metrics.

A Games–Howell post hoc analysis revealed a significant difference for Mental Demand responses between the sound conditions Close and Silence ($p_{\mathrm{GH}}=0.041$, Mean Difference = 18.57, 95% CI [0.69, 36.46], d=0.987) but not between the conditions Close and Far ($p_{\mathrm{GH}}=0.895$, Mean Difference = 4.29, 95% CI [−19.41, 27.98], d=0.170) and Far and Silence ($p_{\mathrm{GH}}=0.218$, Mean Difference = 14.29, 95% CI [−6.61, 35.18], d=0.655).

### 3.3. Continuous Performance Test

#### 3.3.1. Analysis

In order to analyze the impact of the sound condition on the CPT metrics, two repeated measures analyses of variance (rmANOVAs) were individually performed for both the number of commission errors and the mean response time results. Prior to this analysis, it was found that the mean response time results for all three sound conditions are right-skewed; hence, a 1/x transformation was applied. After this transformation, all assumptions for a repeated-measures ANOVA were met.

Figure 7 shows arithmetic means and 95% confidence intervals of both metrics under the three different sound conditions for both male and female participants as well as for the combination of all participants. Thereby, it can be seen that female participants tend to have a lower number of commission errors, which is consistent with normative data for the Conners CPT [34], as well as shorter mean response times than male participants. This impact of gender on CPT performance described in the literature could motivate a gender-specific analysis. However, since both metrics show a very similar trend among the three sound conditions for both male and female participants, it can be assumed that gender did not influence the response to traffic noise in the CPT task; hence, it was decided to conduct the rmANOVAs without including it as a between-subject factor.

In addition to analyzing the group means, it was evaluated for which sound condition each individual participant achieved their lowest and highest number of commission errors. The number of participants with personal minimum and maximum commission errors for all three sound categories was counted as shown in Figure 8. Thereby, cases where a participant achieved the same personal lowest or highest number of errors in multiple sound conditions were split up between the counts for those categories.

#### 3.3.2. Results

The repeated measures ANOVA determined a significant effect of the factor sound condition on the number of commission errors in the CPT test (F(2,82)=3.683, p=0.029, partial $\eta^{2}=0.082$) with a medium effect size [48]. A Bonferroni-adjusted post hoc analysis revealed a significantly higher number of commission errors for the sound condition Far compared to sound condition Silence ($p_{\mathrm{bon}}=0.045$, Mean Difference = 1.571, 95% CI [0.026, 3.117], $d_{z}=0.392$) with a relatively small effect size [48]. No significant difference in the number of commission errors was found between the sound conditions Close and Far ($p_{\mathrm{bon}}=1.000$, Mean Difference = −0.286, 95% CI [−1.878, 1.306], $d_{z}$ = −0.069) as well as between sound conditions Close and Silence ($p_{\mathrm{bon}}=0.108$, Mean Difference = 1.286, 95% CI [−0.193, 2.765], $d_{z}=0.335$). Finally, a contrast analysis showed a significant difference between the combination of commission errors for both noise conditions (Close and Far) compared to Silence (p=0.008, Mean Difference = −1.429, 95% CI [−2.469, −0.388]). These results align with the distribution of individual best and worst performances presented in Figure 8, which show that 19.5 participants achieved their personal best performance, i.e., the lowest number of errors, in Silence and only 8.5 participants had their best performance under the sound condition Close. However, Figure 8a also indicates a difference in the number of participants with individual best performance between the sound conditions Close (N=8.5) and Far (N=14), which is neither apparent in the group means shown in Figure 7a nor in the rmANOVA results.

The repeated measures ANOVA for the mean response time results showed no significant impact of the sound condition (F(2,82)=0.580, p=0.562, partial $\eta^{2}=0.014$), and no further post hoc tests were performed.

### 3.4. Correlation

Table 3 shows Pearson’s correlation coefficients for the NASA-TLX results and the first CPT round. These coefficients show a strong negative correlation between the number of commission errors and the response time in the CPT test. Additionally, a moderate correlation between different dimensions of the NASA-TLX results was found, which has already been reported by the original authors of the test [42]. Furthermore, the results for Performance are moderately positively correlated to the number of commission errors as well as negatively correlated to the mean response time. Since a high reported value on the performance scale is associated with “failure”, this correlation indicates that participants who reported worse performance obtained a higher number of errors while responding faster. Finally, the number of commission errors is weakly correlated with reported Frustration.

In order to investigate a potential relation between the self-reported loudness preference and the performance in the CPT task, the number of commission errors for each participant and both noise conditions was divided by the number of commission errors for the Silence condition. The resulting ratios of numbers of errors while exposed to noise compared to silence showed no significant correlation with the self-reported loudness preference, neither for the Close condition (Pearson’s r=0.077, p=0.629) nor for the Far condition (Pearson’s r=0.044, p=0.782).

## 4. Discussion

This study performed a laboratory experiment in order to investigate the impact of distance-induced temporal changes in road traffic noise signals on perceived workload and cognitive performance. The research hypothesis was that the presence of traffic noise affects both metrics and that nearby traffic noise has a higher negative impact than distant traffic noise with the same equivalent sound pressure level. Additionally, participants answered a simple questionnaire in order to provide a better understanding of the demands for supportive acoustic environments as well as the relation between subjective noise preference and objective performance in the experiments.

### 4.1. Noise Preference

The results from the noise preference questionnaire show that the majority of the evaluated group of participants found speech to be most unfavorable as an acoustic environment for concentrated working, which could be explained by the irrelevant speech effect [49,50]. However, also, loud and unexpected sounds, which might be more similar in time structure to close traffic noise, are perceived as negative, while white noise, which has a time structure more similar to traffic at a far distance, is described as positive by some participants. These subjective evaluations hence could be interpreted to speak in favor of the working hypothesis that the time structure of nearby traffic is perceived as more disturbing than the time structure of distant traffic.

Additionally, it was found that the majority of the evaluated group of participants favors calm music or silence as an acoustic environment for concentrated working. However, the personal interpretation of the term silence might differ between participants. While for some people, silence might refer to the total absence of any sounds, a scenario that is relatively uncommon in everyday life, others might rather refer to a quiet acoustic environment such as a library, which, even though it is referred to as silent, still contains some low-level background noise. This discrepancy becomes clear when comparing the free text questionnaire answers, where 20 participants responded to prefer silence as acoustic working environment, to the results of the preferred loudness scale, where only four participants responded to prefer absolutely quiet environments. This finding reveals a potential problem in the experiment design since, due to the acoustic properties of the lab environment, the Silence sound condition had a very low background noise level of $L_{\mathrm{Aeq}}\approx 10$ dB, which is below the typical background noise level of ordinary silent environments [51]. The preferred loudness results indicate that this condition might have actually been too quiet for the personal preference of most participants.

Even though this difference was found to not be significant, the preferred loudness scale results also indicate that female participants tend to prefer slightly quieter environments than male participants. This trend is consistent with studies such as [52], which found that females tend to higher ratings of noise annoyance and fatigue than males. Finally, the fact that no correlation between the preferred loudness results and the performance in the CPT test was found indicates that simple self-reported questionnaires are not necessarily reliable predictors for the objective human response to noise. However, studies such as [53] found that the interaction between noise sensitivity, as assessed by a more sophisticated Weinstein noise sensitivity scale [54], and noise type significantly affects working memory accuracy, which leads to the assumption that the questionnaire used in this study might not have been specific enough to capture these interactions.

### 4.2. Perceived Task Load

The NASA-TLX results reveal that traffic noise at an indoor level of $L_{\mathrm{Aeq}}\approx 40$ dBA can influence subjective task load and especially the perceived mental demand. While none of the NASA-TLX subscales shows a significant difference between the two traffic noise conditions, the group means of Mental Demand, Physical Demand, Temporal Demand and Frustration decrease from sound condition Close to Far. This trend speaks in favor of our working hypothesis and could indicate that while not significant with the used methods, other conditions such as more pronounced temporal changes, higher noise levels or a larger group of participants might reveal significant differences in subjective workload for changes in the noise time structure. For the metrics Effort and Frustration, the group means for the Silence sound condition were found to be higher than for the Far sound condition. Even though these differences were not statistically significant, this trend could indicate that there are aspects of subjective workload for which extreme silence is perceived as more demanding than moderate background noise. This observation aligns with the results of the self-reported noise preference, which show that the majority of participants do not favor absolute silence for concentrated working and could motivate further research on the impact of “unnatural” silence on subjective metrics such as perceived workload.

### 4.3. Cognitive Performance

The results of the continuous performance test indicate that the evaluated traffic noise at an indoor level of $L_{\mathrm{Aeq}}\approx 40$ dBA negatively affects sustained attention and inhibitory control. This is a relevant finding, since, to our best knowledge, an effect of such relatively low traffic noise levels on cognitive performance has hardly been shown yet. This could mean that the implemented CPT method is more sensitive to distraction by traffic noise than other commonly used paradigms, which somewhat contradicts the results of Ballard [39], who found that both constant and fluctuating white noise at a level of 90 dB does not influence performance in a similar response-inhibition CPT task. However, one could argue that due to its time and frequency structure, complex traffic noise might be perceived as more distracting than the white noise used by Ballard. Alternatively, the low background noise level of the lab and the circumstance that heavy vehicle recordings with more low-frequency content than regular cars were used could contribute to this observed effect at moderate noise levels. All those assumptions underline that reporting only time-averaged single values such as the $L_{\mathrm{Aeq}}$ might not be sufficient to predict the human response to traffic noise.

Even though a visual inspection of the results, as well as normative data for the Conners CPT test, indicates an effect of gender on the overall number of commission errors [34], it was found that the presence of traffic noise had a very similar impact on the performance of both female and male participants. However, this assumption is based on a simple descriptive data analysis and could be statistically confirmed in a follow-up study with a gender-balanced group of subjects.

The main research question of this study was whether distance-induced temporal changes in traffic noise affect cognitive performance. While the distribution of the lowest number of errors presented in Figure 8a seems to support our research hypothesis that the Close noise has a higher impact on cognitive performance than the Far noise, none of the performed statistical analyses confirms that this difference is actually significant, which aligns with the findings of Schaeffer et al. [26]. This could either indicate that the time structure does not play a major role in the human response to traffic noise at moderate levels or that the used methods are not sensitive enough to detect a possible effect. However, the implemented CPT test primarily measures sustained attention and inhibitory control, and since Lavandier et al. [28] found an influence of time structure on reading task performance, there is reason to assume that other aspects of cognitive performance might be affected.

### 4.4. Outlook

While the noise levels chosen for this study are comparably low for a laboratory experiment, they are still above regulations such as the Swedish building code [55] which, in combination with the relatively low background noise, means that the evaluated scenarios are not necessarily representative for real road-traffic noise in urban environments. Even though it might seem counter-intuitive, we suspect that lower traffic noise levels in combination with increased background noise could actually reveal more differences between the time structures for cases where only the peaks of the Close traffic noise become audible and the less dynamic Far signal is completely masked by background noise. This assumption could be investigated in a follow-up experiment with decreased traffic noise levels and additional background noise, which is more adequate for an office or residential environment.

An additional factor that has not been included in this study is the physical limit for low-frequency façade insulation when building houses close to streets. We assumed that the increased equivalent sound pressure level due to a reduced street distance is frequency independent compensated by an improved façade construction; hence, both the Close and the Far stimuli have the same $L_{\mathrm{Aeq}}$ and frequency composition. In real life, it would be difficult to increase the façade insulation by, in our case, approximately 7 dB equally for all frequencies. Instead, the Close indoor noise would most likely contain more energy at low frequencies which, due to the applied A-Weighting, only has a minor impact on the resulting $L_{\mathrm{Aeq}}$. Since studies such as [56] showed that low-frequency noise can negatively affect cognitive performance compared to reference noise with less low-frequency energy but the same $L_{\mathrm{Aeq}}$, there is reason to assume that in a real-life scenario, differences between nearby and distant traffic noise with the same $L_{\mathrm{Aeq}}$ would have a larger effect than the results of our study indicate. Conducting an experiment that includes both temporal and spectral changes would be valuable in order to better understand the impact of road-to-façade distance on the human response to traffic noise.

## 5. Conclusions

This study investigated the effect of distance-induced temporal changes on the human response to road traffic noise by performing a laboratory listening experiment comparing three different sound conditions (Close and Far traffic noise, both with different time structures but same $L_{\mathrm{Aeq}}\approx 40$ dB, and Silence). Thereby, the perceived task load was evaluated using a NASA-TLX test, and cognitive performance was measured using a response-inhibition continuous performance test. The results show a significant effect of the sound condition on the perceived task load as assessed by a multivariate analysis of variance (F(10,70)=2.394, p=0.017, partial $\eta^{2}=0.255$, Wilk’s Λ=0.555) as well as on the number of commission errors in the continuous performance test as assessed by a repeated measures analysis of variance (F(2,82)=3.683, p=0.029, partial $\eta^{2}=0.082$). While some of the results show a trend of decreasing noise impact when increasing the traffic distance, post hoc tests only showed significant differences between noise and Silence but not between the sound conditions Close and Far. This indicates that traffic noise at $L_{\mathrm{Aeq}}\approx 40$ dB can influence both perceived task load and cognitive performance and that time structure might only play a minor role in the human response. If distance-induced temporal changes in traffic noise cause differences in cognitive performance, the used methods are not sufficiently sensitive to detect them.

## Figures and Tables

**Figure 1 ijerph-20-03798-f001:**
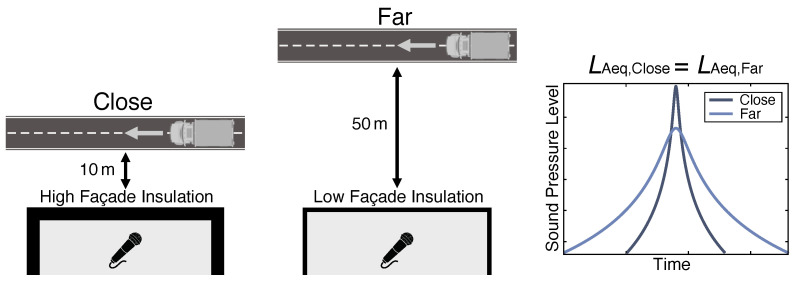
Visualization of the relation between source distance and indoor noise time structure for a typical road-traffic scenario with adapted façade insulation. Even though the equivalent indoor sound pressure levels for both cases are exactly the same, the time structure of the two signals is different since the maximum sound pressure level $L_{\mathrm{AF},\max}$ decreases by ≈6 dB per distance doubling while the time-averaged sound pressure level $L_{\mathrm{Aeq}}$ only decreases by ≈3 dB.

**Figure 2 ijerph-20-03798-f002:**
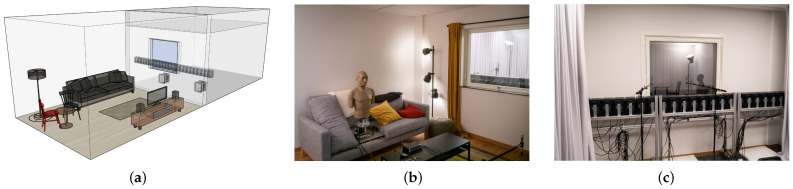
Model of Living Room Lab (**a**), listening room with artificial head at participant position (**b**) and loudspeaker array setup behind window in transmitting room (**c**).

**Figure 3 ijerph-20-03798-f003:**
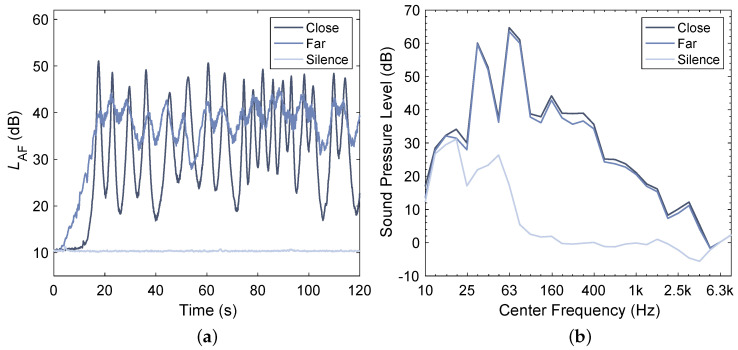
Time structure excerpt (**a**) and third-octave band levels (**b**) of acoustic stimuli recorded at participant position.

**Figure 4 ijerph-20-03798-f004:**
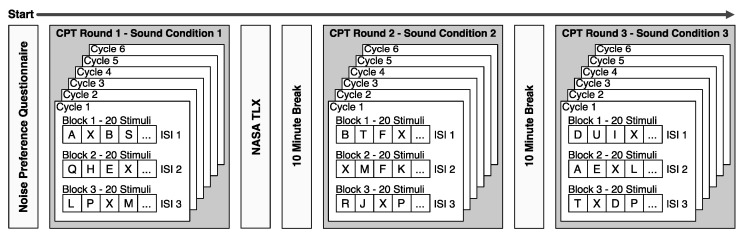
Overall experiment procedure.

**Figure 5 ijerph-20-03798-f005:**
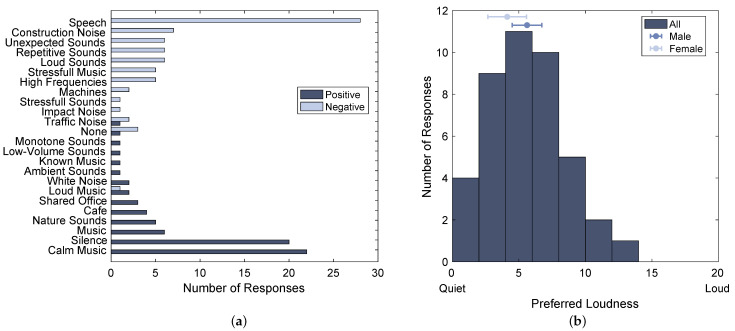
Participants’ responses of (**a**) positive and negative acoustic environments as well as (**b**) loudness preference for concentrated working. Error bars indicate mean and 95% confidence intervals for male and female participants.

**Figure 6 ijerph-20-03798-f006:**
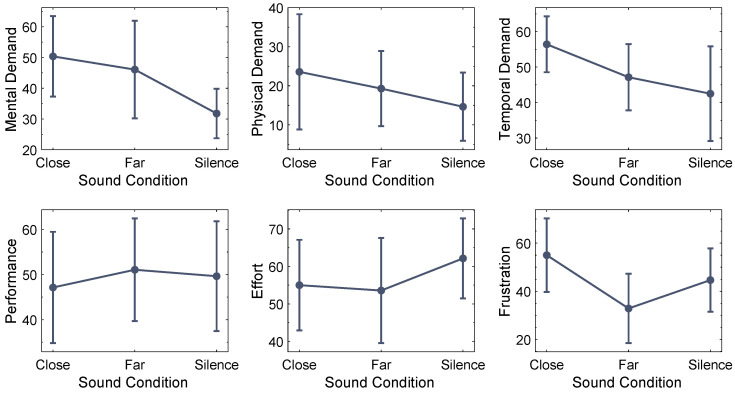
Means and 95% confidence intervals of perceived task load results.

**Figure 7 ijerph-20-03798-f007:**
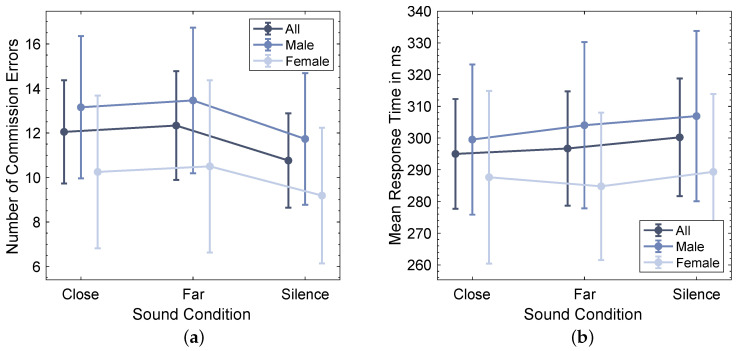
Means and 95% confidence intervals of number of commission errors (**a**) and mean response time (**b**) in continuous performance test.

**Figure 8 ijerph-20-03798-f008:**
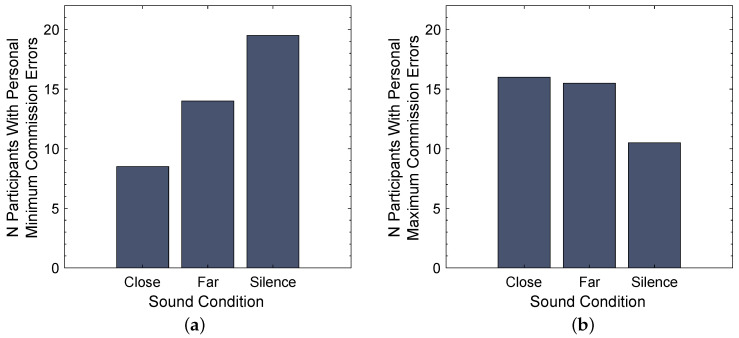
Number of participants with personal minimum (**a**) and maximum (**b**) number of commission errors for each sound condition.

**Table 1 ijerph-20-03798-t001:** Properties of stimuli, sound pressure levels obtained from recordings at participant position.

Stimulus	Road–Façade Distance	Vehicles/h	Average Speed	Indoor *L*_Aeq_	Indoor *L*_AF,max_
Close	10 m	600	50 km/h	40.4 dB	51.7 dB
Far	50 m	600	50 km/h	39.1 dB	46.7 dB
Silence	-	-	-	10.4 dB	12.2 dB

**Table 2 ijerph-20-03798-t002:** Post hoc Welch ANOVA results for NASA-TLX Test.

Variable	df	Welch’s *F*	*p*	${\hat{\omega }}^{2}$
Mental Demand	(2,23.882)	3.958	0.033	0.123
Temporal Demand	(2,25.061)	2.390	0.112	0.062
Performance	(2,25.967)	0.126	0.882	−0.043
Effort	(2,25.691)	0.705	0.503	−0.014
Frustration	(2,25.902)	2.546	0.098	0.069

**Table 3 ijerph-20-03798-t003:** Pearson’s correlation coefficients for first round CPT and NASA TLX results.

Variable	1	2	3	4	5	6	7	8
1. Mental Demand	–							
2. Physical Demand	0.37 *	–						
3. Temporal Demand	0.59 **	0.37 *	–					
4. Performance	0.19	0.17	−0.07	–				
5. Effort	0.46 **	0.24	0.40 **	0.27	–			
6. Frustration	0.38 *	0.37 *	0.27	0.39 *	0.53 *	–		
7. Commission Errors	0.12	0.09	0.08	0.38 *	0.34 *	0.29	–	
8. Mean Response Time	0.00	0.10	0.07	−0.33 *	−0.18	−0.19	−0.65 **	–

* *p* < 0.05. ** *p* < 0.01.

## Data Availability

The data presented in this study as well as binaural stimuli recordings are openly available in https://doi.org/10.5281/zenodo.7565815 (accessed on 25 January 2023).

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
