# Peer review of "Traffic Noise at Moderate Levels Affects Cognitive Performance: Do Distance-Induced Temporal Changes Matter?"

_ijerph, 2023, doi:10.3390/ijerph20053798_

Round 1

Reviewer 1 Report

This paper addresses a subtle question: if the traffic noise with a different temporal distribution due to major distance, equalized in level to the noise coming from a closer source had a different effect on cognitive functions and perceived workload.

The described experiment is methodologically correct and the paper is well written.

The authors are aware that the experimental design could be improved. For example they are conscious that the “silence” condition they set corresponds to an unreasonably low sound level. Such a low level could be associated to anxiety much more than to relax and easiness in concentration. This explain the some paradoxical results shown in Fig.6. The hypothesis of this study was that the traffic noise  coming from a closer source, also if the façade insulation had been increased and, consequently, the noise level had been equalized to that of a farer source, should cause a decline in cognitive performance and an increase in the perceived workload. On this respect, this paper did not find significant differences between the close or far source. Or, in other words, this paper did not find a significant effect of the different time distribution of the traffic noise in the case of “Close” or “Far” source.

The authors used the MANOVA to analyze the data of NASA-TLX test variables, the six variables: Mental Demand, Physical Demand, Temporal Demand, Performance, Effort, Frustration. I am not convinced that this approach is definitely correct. In fact, this study design is based on measures repeated on the same subject. The authors used indeed the ANOVA for repeated measurements in the case  of the factor sound condition on the number of commission errors in the CPT test. I would suggest to use the repeated measurements statistical test approach also in the case of the NASA test or, if the authors do not think this would be the correct approach, they should explain to me and to the reader the reason.  I would suggest to use the mixed effect linear regression models, in which the subject can be added as a random variable.

You could arrange the “Mental Demand”, “Physical Demand” etc in a unique vector (I call it “performance”), labelling the kind of performance and creating a factor variable. The model should explain performance with the three level sound condition. The model is written in the following:

lme( performance ~sound_cond*kind , random = ~1|subj)

subj represents the random variable subject

Author Response

Dear Reviewer,

thank you for your input! Please find our responses in the attachment.

Reviewer 2 Report

First of all, as the reviewer of this paper, I would like to offer my sincere congratulations on the work of the authors of this article. In my opinion, this would be an outstanding piece of work.

The clarity of the explanation, as well as the transparency in providing the data through open access links and the use of a large number of references to current works, make this publication a very complete and high-quality document.

As a single comment, in my opinion, the "Conclusions" section should be extended with a short summary of the most relevant points of the work, providing references to the data presented in the article.

Author Response

(The authors gave the same response as above.)

Reviewer 3 Report

The submitted document reports an excellent work investigating reaction to indoor noise. In my opinion, the paper is almost close to publication. My few comments are reported below.

Avoid figures in the introduction.

In environmental acoustics, 1 decimal digit is enough.

Health effect parts in the introduction is too brief and not complete. I suggest the authors to spend a couple more lines and add more effects/reference. A suggestion can be:

 Exposure to noise is associated to sleep disorders with awakenings (Muzet A. Environmental noise, sleep and health. Sleep Med Rev 2007; 11: 135–42), learning impairment (Zacarías, F. F., Molina, R. H., Ancela, J. L. C., López, S. L., & Ojembarrena, A. A. (2013). Noise exposure in preterm infants treated with respiratory support using neonatal helmets. Acta Acustica united with Acustica, 99(4), 590-597; Minichilli, Fabrizio, et al. "Annoyance judgment and measurements of environmental noise: A focus on Italian secondary schools." International journal of environmental research and public health 15.2 (2018): 208; Erickson, Lucy C., and Rochelle S. Newman. "Influences of background noise on infants and children." Current Directions in Psychological Science 26.5 (2017): 451-457.), hypertension ischemic heart disease (Dratva, J., et al. (2012). “Transportation noise and blood pressure in a population‐based sample of adults.” Environmental Health Perspectives, 120(1): 50–55. Babisch, W., Beule, B., Schust, M., Kersten, N., Ising, H., ‘Traffic noise and risk of myocardial infarction’, Epidemiology, 16, 2005, pp. 33–40. ), diastolic blood pressure (Petri, D., et al. (2021). Effects of Exposure to Road, Railway, Airport and Recreational Noise on Blood Pressure and Hypertension. Int. J. Environ. Res. Public Health 2021, 18(17), 9145), reduction of working performance (Vukić, L., et al. (2021). Seafarers’ Perception and Attitudes towards Noise Emission on Board Ships. International Journal of Environmental Research and Public Health, 18(12), 6671. Rossi, L., Prato, A., Lesina, L., & Schiavi, A. (2018). Effects of low-frequency noise on human cognitive performances in laboratory. Building Acoustics, 25(1), 17-33.), annoyance (Miedema HME, Oudshoorn CGM. Annoyance from transportation noise: relationships with exposure metrics DNL and DENL and their confidence intervals. Environ Health Perspect 2001; 109: 409–16; Licitra, G., et al. (2016). Annoyance evaluation due to overall railway noise and vibration in Pisa urban areas. Science of the total environment, 568, 1315-1325.).”

Author Response

(The authors gave the same response as above.)
